# Role of Porcine Aminopeptidase N and Sialic Acids in Porcine Coronavirus Infections in Primary Porcine Enterocytes

**DOI:** 10.3390/v12040402

**Published:** 2020-04-05

**Authors:** Tingting Cui, Sebastiaan Theuns, Jiexiong Xie, Wim Van den Broeck, Hans J. Nauwynck

**Affiliations:** 1Laboratory of Virology, Faculty of Veterinary Medicine, Ghent University, Salisburylaan 133, B-9820 Merelbeke, Belgium; tingting.cui@ugent.be (T.C.); sebastiaan.theuns@ugent.be (S.T.); jiexiong.xie@ugent.be (J.X.); 2Department of Morphology, Faculty of Veterinary Medicine, Ghent University, Salisburylaan 133, B-9820 Merelbeke, Belgium; wim.vandenbroeck@ugent.be

**Keywords:** enterocytes, porcine coronavirus, transmissible gastroenteritis virus (TGEV), Porcine epidemic diarrhea virus (PEDV), aminopeptidase N, sialic acids

## Abstract

Porcine epidemic diarrhea virus (PEDV) and transmissible gastroenteritis virus (TGEV) have been reported to use aminopeptidase N (APN) as a cellular receptor. Recently, the role of APN as a receptor for PEDV has been questioned. In our study, the role of APN in PEDV and TGEV infections was studied in primary porcine enterocytes. After seven days of cultivation, 89% of enterocytes presented microvilli and showed a two- to five-fold higher susceptibility to PEDV and TGEV. A significant increase of PEDV and TGEV infection was correlated with a higher expression of APN, which was indicative that APN plays an important role in porcine coronavirus infections. However, PEDV and TGEV infected both APN positive and negative enterocytes. PEDV and TGEV Miller showed a higher infectivity in APN positive cells than in APN negative cells. In contrast, TGEV Purdue replicated better in APN negative cells. These results show that an additional receptor exists, different from APN for porcine coronaviruses. Subsequently, treatment of enterocytes with neuraminidase (NA) had no effect on infection efficiency of TGEV, implying that terminal cellular sialic acids (SAs) are no receptor determinants for TGEV. Treatment of TGEV with NA significantly enhanced the infection which shows that TGEV is masked by SAs.

## 1. Introduction

Coronaviruses are known as human and animal pathogens that mainly infect the epithelium of the respiratory or intestinal tract. Porcine epidemic diarrhea virus (PEDV), transmissible gastroenteritis virus (TGEV), and its variant porcine respiratory coronavirus (PRCV) are classified as Alphacoronavirus. They are enveloped viruses containing a single-stranded, positive-sense RNA genome of approximately 28.5 kb. The positive ssRNA serves as mRNA for the generation of viral replicative proteins by translation of open reading frame (ORF) 1a and ORF1b. The genome contains a 5′ untranslated region (UTR), a 3′ UTR, and at least seven ORFs. ORF1a and ORF1b make up two-thirds of the viral genome and encode the non-structural replicase polyproteins (replicases 1a and 1b), which further guide the viral replication and translation, regulate cellular processes, and potentially fulfill other unknown functions. The remaining proximal third of the genome encodes four structural proteins (spike (S), envelope (E), membrane (M), and nucleocapsid (N) by ORF2, ORF4, ORF5, and ORF6, respectively). The S protein is a type I glycoprotein that projects from the virions surface forming the corona-like appearance of viral particles. It contains an N-terminal ectodomain (S1) and a short C-terminal cytoplasmic tail (S2). It is known that the peripheral S1 portion is a globular cellular receptor binding subunit while the transmembrane S2 portion is required to mediate fusion of viral and cellular membranes [1]. The S protein also contains important antigenic determinants for coronaviruses. The triple-spanning M protein has a short N-terminal glycosylated domain, a long C-terminal domain, and three transmembrane domains [2]. M proteins play key roles in virion assembly/budding. The lateral interactions between M and E proteins are supposed to mediate the formation of the virion envelope. The M–S protein interactions are also needed to retain the spike proteins at the budding site, incorporating the S proteins into the viral envelope [3]. The E protein is a small hydrophobic protein of 9–12 kDa. It plays crucial roles during virus budding and transiently localizes at the pre-Golgi compartment before progressing to the Golgi apparatus. Deletion of the E protein in TGEV arrests virus transportation and maturation [4]. The E protein possesses an ion channel activity, contributing to virus virulence and pathogenesis [5].

The replication of viruses in their host cells is firstly mediated by the virus binding to the cellular receptors. Some Alphacoronaviruses, including human coronavirus 229E (HCoV-229E), feline infectious peritonitis virus (FIPV), canine coronavirus (CCoV), TGEV, and PEDV, may use aminopeptidase N (APN) as a cellular receptor [6,7,8,9,10]. APN is a type II cell membrane metalloprotease, that is linked to cells by its N-terminal cytoplasmic domain. It is highly abundant on the brush border membrane of the mature small intestinal enterocytes, kidney, and liver epithelial cells. APN is highly species specific. Human coronaviruses cannot recognize porcine APN as a cellular receptor because of certain differences in glycosylation regions [11]. APN has been identified as a major receptor for TGEV, as it has been shown that TGEV virions are able to specifically bind to purified APN and conferred the infectivity of TGEV to a non-permissive cell line when the cells were expressing recombinant APN [7]. An interaction between APN and PEDV infection has also been demonstrated. Transient expression of APN in Mardin-Darby canine kidney (MDCK) cells conferred susceptibility of these cells to PEDV infection and the infection was inhibited by anti-APN polyclonal antibodies [9]. Cong and colleagues demonstrated that APN mediates PEDV infection in Vero E6 and in an immortalized porcine enteric cell line [12]. However, increasingly more data demonstrates that APN is not essential for PEDV infection, because PEDV exhibits no binding to soluble APN [13,14]. Moreover, Vero cells are commonly used for PEDV propagation and cultivation in vitro. The endogenous expression of APN in Vero cells was undetectable in the mRNA and the protein levels, which challenges the role of APN as a cellular receptor for PEDV. Therefore, whether APN is a functional receptor for PEDV is disputable. However, no data are available on the role of APN in primary enterocytes.

In addition to APN, a sialic acid binding capacity has been reported for TGEV and PEDV [15,16]. Sialic acids are electronegatively charged monosaccharides in animals and some microorganisms. They are prominently positioned at the outer end of *N*-glycans, *O*-glycans, and glycosphingolipids. TGEV shows a hemagglutinating activity with α-2,3-linked sialic acids on the erythrocyte surface as target. Among different types of sialic acids, *N*-glycolylneuraminic acid was recognized by TGEV more efficiently than *N*-acetylneuraminic acid [15]. Sialic acid binding by TGEV is not essential in the initiation of infection of cultured cells, as TGEV mutants lacking sialic acid binding activity can be propagated to the same extent in cultured cells [15]. However, sialic acid binding activity is important for the enteropathogenicity of TGEV. Mutants without sialic acid binding were not capable of initiating intestinal infection in vivo, showing that sialic acid binding is required for efficient intestinal infections [17]. This sialic acid binding activity of TGEV is supposed to help the virus to penetrate the mucus layer and to infect intestinal epithelial cells. PEDV is also able to bind sialic acids. PEDV S1 protein was reported to bind bovine and porcine mucins that contain a mixture of different types of sugar [16,18]. Furthermore, Neu5Ac was identified as the most favored binding sugar of PEDV S1 by a glycan array screen [16]. Different from the APN binding domain in the C-terminal of PEDV S1 (residues 477–629), the sialic acid binding domain was reported in the N-terminal of PEDV S1 (residues 1–320) [18].

In the present study, the co-culture system of primary porcine enterocytes that was established in our laboratory was used to study the role of APN in PEDV and TGEV infection in their target cells [19]. Further, it was investigated whether sialic acids are cellular receptors for TGEV in primary enterocytes.

## 2. Materials and Methods

### 2.1. Cells, Viruses, and Reagents

Primary porcine enterocytes were isolated from the ileum of three-day-old piglets and co-cultured with porcine myofibroblasts [19]. Euthanizing piglets was done in agreement with the European legislation on animal experiments. All experimental procedures were approved by the Local Ethical Committee of the Faculty of Veterinary Medicine, Ghent University (EC2013/97), and all methods were carried out in accordance with the approved guidelines. The enterocytes were maintained with Dulbecco’s modified Eagle’s-F12 Ham medium (DMEM-F12). TGEV Purdue and Miller grown on swine testicle (ST) cells and PEDV CV777 strain grown on Vero cells were used in this study. PEDV CV777 fecal suspension was collected from a three-day-old infected suckling piglet. A twenty percent fecal suspension was prepared in phosphate buffered saline (PBS) containing 1000 U/mL penicillin (Continental Pharma, Puurs, Belgium), 1 mg/mL streptomycin (Certa, Braine l’Alleud, Belgium), 1 mg/mL gentamicin (Gibco BRL, Merelbeke, Belgium), and 0.01% *v*/*v* fungizone (Bristol-Myers Squibb, Braine l’Alleud, Belgium). Hydrocortisone, spermidine, and Wnt agonist were purchased from Sigma-Aldrich (Sigma, St-Louis, Mo, USA). Porcine insulin was purchased from Protein specialists (ProSpec, Rehovot, Israel). The small intestinal contents (IC) were collected from the duodenum of a three-day-old suckling piglet. After euthanasia, a 20 cm long segment of duodenum was closed by two surgical clamps and was removed from a piglet. Then, one clamp was removed and the intestinal contents were released from the lumen into a 15 mL centrifugation tube. In order to collect all the intestinal contents, the lumen was washed once by filling it with 5 mL Dulbecco’s modified Eagle’s medium (DMEM) containing 1000 U/mL penicillin, 1 mg/mL streptomycin, 1 mg/mL gentamicin, and 0.01% *v*/*v* fungizone. Then, the DMEM was released from the lumen into the 15 mL tube that already contained the intestinal contents. After centrifugation (1200 rpm, 10 min at 4 °C), the supernatant of the intestinal contents was collected and stored at −20 °C until use.

### 2.2. Scanning Electron Microscopy

Three- and seven-day-old primary porcine enterocytes were fixed in HEPES-buffered glutaric aldehyde (Sigma, St-Louis, Mo, USA) for scanning electron microscopy as described previously [20]. After 24 h fixation, the samples were treated with 1% osmium tetroxide for 2 h at room temperature (RT), followed by ascending grades of alcohol dehydration. In order to avoid the water vaporization obstructing the electron beam and interfering with image clarity, the dehydrated samples were transferred to a critical point drier (CPD, Bal-tec, Balzers, Liechtenstein) for complete drying. Finally, the dried samples were mounted on a metal stub and were sputter-coated with platinum. The microvilli of all the samples were acquired with a JEOL JSM 5600 LV scanning electron microscope (JEOL Ltd., Tokyo, Japan).

### 2.3. Treatment of Primary Enterocytes with Hydrocortisone, Spermidine, Porcine Insulin, Wnt Agonist, or Small Intestinal Contents

Twenty-four h post co-cultivation, monolayers of enterocytes were cultured with medium containing hydrocortisone (1 and 10 µg/mL), spermidine (50 and 500 µM), insulin (1 and 10µg/mL), Wnt agonist (0.1 µM), or intestinal contents (1%) for 24 h. The cells were fixed with 4% paraformaldehyde for 10 min and immunofluorescence staining was conducted for APN expression analysis.

### 2.4. Immunofluorescence Staining of the APN Expression in Primary Enterocytes

Cells were incubated with mouse monoclonal anti-porcine APN antibodies (IMM013; kindly provided by Prof. Eric Cox, Ghent University) containing 10% normal goat serum for 1 h at 37 °C, followed by goat anti-mouse-IgG FITC labeled antibodies for 1 h at 37 °C. Nuclei were stained with Hoechst for 10 min at RT. The percentage of APN positive cells were analyzed by fluorescence microscopy (Leica Microsystems GmbH).

### 2.5. Infection Experiment

At twenty-four h of co-cultivation, monolayers of enterocytes were treated with 1 µg/mL hydrocortisone, 50 µM spermidine, 1 µg/mL insulin, 0.1 µM Wnt agonist, or 1% intestinal contents for another 24 h. Then, the susceptibility of treated enterocytes to PEDV and TGEV was tested. Cells were inoculated with 200 µL of TGEV Miller at a multiplicity of infection (m.o.i.) of 1 and 200 µL of the PEDV CV777 Vero adapted strain at 10^5.6^ TCID_50_/mL or 10^7^ viral RNA copies/mL of fecal suspension with 10 µg/mL trypsin. After 60 min of incubation at 37 °C, unbound viral particles were removed by three washing steps with DMEM. Cells were further incubated in medium containing 1 µg/mL hydrocortisone, 50 µM spermidine, 1 µg/mL insulin, 0.1 µM Wnt agonist, or 1% intestinal contents for 24 h (37 °C, 5% CO_2_) and fixed with 4% paraformaldehyde for immunofluorescence staining.

### 2.6. Co-localization Analysis of Viral Antigens with APN by Confocal Microscopy

To determine the co-localization of viral antigens and APN, co-cultured enterocytes were infected with TGEV Miller and Purdue and PEDV Vero adapted and non-adapted strains. After 24 h incubation, cells were fixed with 4% paraformaldehyde for 10 min and immunofluorescence staining was performed. Cells were incubated with mouse monoclonal anti-porcine APN antibodies containing 10% normal goat serum for 1 h at 37 °C, followed by goat anti-mouse-IgG1 FITC labeled antibodies for 1 h at 37 °C. Then, cells were permeabilized with 0.1% Triton X-100 for 5 min at RT and incubated with mouse monoclonal anti-porcine PEDV antibodies (kindly provided by Prof. Luis Enjuanes, National Center for Biotechnology) or swine polyclonal anti-TGEV antibodies [19] containing 10% normal goat serum for 1 h at 37 °C, followed by goat anti-mouse IgG2a AF594 labeled antibodies or goat anti-swine Texas Red labeled antibodies (Molecular Probes). Nuclei were stained with Hoechst for 10 min at RT and the results were analyzed by a Leica TCS SPE laser scanning spectral confocal system linked to a DM B fluorescence microscope (Leica Microsystems).

### 2.7. Neuraminidase Treatment of Cells and Virus

To remove SAs from enterocytes, monolayers of co-cultured enterocytes were washed three times with warm DMEM. Then, cells were incubated with 50 mU/mL NA from Vibrio Cholerae (Roche Diagnostics, Risch-Rotkreuz, Switzerland) at 37 °C for 1 h. Cells that were mock-treated were incubated with DMEM and underwent the same manipulations as NA-treated cells. To remove SAs from the virus, virus suspensions were incubated with 50 mU/mL NA from Vibrio Cholerae at 37 °C for 1 h. the mock-treated virus was incubated in DMEM and underwent the same manipulations as the NA-treated virus. Afterwards, cells were inoculated with either NA-treated or mock-treated virus (m.o.i of 1 for TGEV Purdue and Miller). After 60 min incubation at 37 °C, cells were washed three times with medium and further incubated in medium for 24 h (37 °C, 5% CO_2_). Then, cells were fixed with 4% paraformaldehyde for 10 min at RT. Immunofluorescence was performed to measure the percentage of infected cells. The cells were permeabilized with 0.1% Triton X-100 for 5 min at RT. Then, cells were incubated with swine polyclonal anti-TGEV antibodies containing 10% normal goat serum for 1 h at 37 °C, followed by goat anti-swine-IgG FITC labelled antibodies for 1 h at 37 °C. Nuclei were stained with Hoechst for 10 min at RT. The percentages of infected cells were determined by fluorescence microscopy.

### 2.8. Co-localization of TGEV and SAs

To determine the co-localization of viral antigens and SAs, co-cultured enterocytes were infected with TGEV Miller and Purdue. After 24 h incubation, cells were fixed with 4% paraformaldehyde for 10 min and a double immunofluorescence staining was performed. Cells were incubated with biotinylated Maackia amurensis lectin II (Vector laboratories) for 1 h at 37 °C. The lectin was subsequently stained with Streptavidin-FITC (Invitrogen) for 1 h at 37 °C. Then, cells were permeabilized with 0.1% Triton X-100 for 5 min at RT and incubated with swine polyclonal anti-TGEV antibodies containing 10% normal goat serum for 1 h at 37 °C, followed by goat anti-swine-IgG Texas Red labeled antibodies. Nuclei were stained with Hoechst for 10 min at RT and results were analyzed by a Leica TCS SPE laser scanning spectral confocal system linked to a DM B fluorescence microscope.

### 2.9. Binding Assays

To characterize the attachment of TGEV to primary enterocytes, direct virus-binding studies were carried out with TGEV particles. Cells were chilled on ice for 5 min and washed three times with DMEM. Then, cells were inoculated with TGEV Miller and Purdue particles at a m.o.i. of 10 for 1 h at 4 °C. Unbound virus particles were removed by three washings with DMEM. Cells were fixed with 4% paraformaldehyde for 10 min and a double immunofluorescence staining was performed. Cells were incubated with biotinylated Maackia amurensis lectin II or mouse monoclonal anti-porcine APN antibodies for 1 h at 37 °C, followed by Streptavidin-FITC or goat anti-mouse-IgG1 FITC labeled antibodies for 1 h at 37 °C. Then, cells were permeabilized with 0.1% Triton X-100 for 5 min at RT and incubated with swine polyclonal anti TGEV antibodies containing 10% goat serum for 1 h at 37 °C, followed by goat anti-swine-IgG Texas Red labeled antibodies. Nuclei were stained with Hoechst for 10 min at RT and the results were analyzed by a Leica TCS SPE laser scanning spectral confocal system linked to a DM B fluorescence microscope.

### 2.10. Statistical Analysis

Data were statistically processed by SPSS (*t*-test). The data are represented as means with standard deviation (SD) of three independent experiments. Results with *p*-values of <0.05 were considered significant.

## 3. Results

### 3.1. Aged Enterocytes Carry More Microvilli, Express More APN, and Demonstrate Increased Susceptibility to TGEV and PEDV Infection

The percentages of microvilli positive enterocytes were analyzed by scanning electron microscopy. A higher percentage of microvilli positive cells (89%) was observed at seven days cultivation compared to three days cultivation (66%). The expression of APN at seven days cultivation (28.7 ± 2.3%) was significantly higher than at three days cultivation (14.8 ± 3.2%; Figure 1A,B). The data suggest that primary enterocytes underwent a differentiation process in vitro. They terminally differentiate into mature enterocytes during the long cultivation time. Next, enterocytes were inoculated with TGEV and PEDV at three and seven days of cultivation. The results showed that a significantly higher infection was detected in enterocytes at seven days cultivation (Miller: 3.6 ± 1.1%; Purdue: 7.3 ± 0.7%) than at three days cultivation (Miller: 0.7 ± 0.7%; Purdue: 3.5 ± 0.5%) for TGEV. An increased trend of infection was detected in enterocytes at seven days cultivation (39 infected cells per well) than at three days cultivation (16 infected cells per well) for PEDV but without significance (*p* = 0.06; Figure 1C).

### 3.2. Effects of Enterocyte Differentiation Factors (Hydrocortisone, Spermidine, Porcine Insulin, Wnt Agonist, or Small Intestinal Contents) on APN Expression

Cells were treated with hydrocortisone, spermidine, porcine insulin, Wnt agonist, or small intestinal contents to analyse their effects on enterocyte differentiation. Cells were treated with the aforementioned products for 24 h. Afterwards, the differentiation marker APN was stained by immunofluorescence and the percentage of APN positive cells was counted (Figure 2A). The results showed that without treatment, 13.8 ± 2.6% of cells were APN positive. The treatment with 1 and 10 µg/mL hydrocortisone significantly increased the APN expression to 25 ± 5.5% and 25 ± 8.8%, respectively. The treatment with 50 mM and 500 mM spermidine significantly enhanced the APN expression to 27 ± 8.1% and 26 ± 5.8%, respectively. Similarly, 1 and 10 µg/mL insulin treatment significantly increased the APN expression to 25 ± 6.4% and 27 ± 10.1%, respectively. Since there was no dose-dependent enhancement, the lower concentration of each product (1 µg/mL of hydrocortisone, 50 µM of spermidine and 1 µg/mL of porcine insulin) was used for the next experiment. The treatment of Wnt agonist and intestinal contents showed a trend of increased APN expression up to 22 ± 7% (*p* = 0.1) and 22 ± 9.9% (*p* = 0.3), respectively (Figure 2B).

### 3.3. Treatment With Differentiation Factors (Hydrocortisone, Spermidine, Porcine Insulin, Wnt Agonist, or Small Intestinal Contents) Promotes PEDV and TGEV Infections

To determine the effect of APN on coronavirus replication, the enterocytes were precultured with 1µg/mL hydrocortisone, 50 µM spermidine, 1 µg/mL insulin, 0.1 µM Wnt agonist, or 1% intestinal contents for 24 h prior to inoculation with PEDV CV777 Vero adapted strain, CV777 fecal suspension, and TGEV Miller. For PEDV CV777 Vero adapted strain, only 30 ± 11 cells were infected per well without pretreatment. The highest infection (169 ± 81 infected cells per well) was observed in cells that were pretreated with intestinal contents. Because of the variation between the three replicates, the treatment with intestinal contents was not significantly different from the mock treatment (*p* = 0.09). When cells were pretreated with Wnt agonist, a significant increase of infection (87 ± 23 infected cells per well) was observed (*p* = 0.018). An increased trend of infection (but not significantly different) was observed after pretreatment with hydrocortisone (42 ± 34 infected cells per well, *(p* = 0.6), spermidine (50 ± 6 infected cells per well, *p* = 0.052), and porcine insulin (46 ±27 infected cells per well, *p* = 0.4). For CV777 fecal suspension, the highest infection was observed when cells were pretreated with the Wnt agonist (12 ± 7 infected cells per well), showing a four-fold higher infection compared to mock-treated cells (3 ± 4 infected cells per well). When cells were pretreated with intestinal contents, 9 ± 6 cells were infected. For TGEV Miller, incubation with intestinal contents, porcine insulin, and Wnt agonist significantly increased the virus infection from 1.0 ± 0.3% to 3.6 ± 0.9%, 3.2 ± 0.9% and 3.0 ± 0.5%, respectively. Hydrocortisone and spermidine increased the virus infection to 2.7 ± 0.5% and 2.5 ± 0.6% (Figure 3). The results show that pretreatment of primary enterocytes with hydrocortisone, spermidine, porcine insulin, Wnt agonist, and intestinal contents could stimulate the expression of APN and enhance the infection of PEDV CV777 Vero adapted and non-adapted strains and the TGEV Miller in the enterocytes.

### 3.4. PEDV and TGEV Infect Both APN Positive and Negative Enterocytes

To assess the role of APN in the replication of PEDV and TGEV, a double immunofluorescence staining of APN and virus was performed. For CV777 Vero adapted strain, 0.040 ± 0.002% of APN positive cells and 0.020 ± 0.006% of APN negative cells were infected. For CV777 fecal suspension, 0.02 ± 0.02% of APN positive cells and 0.003 ± 0.001% of APN negative cells were infected. For TGEV Miller, 4.0 ± 2.6% of APN positive cells and 1.1 ± 0 3% of APN negative cells were infected. For TGEV Purdue, more infection was found in APN negative cells (3.5 ± 0.6%) than in APN positive cells (2.0 ± 1.2%; Figure 4). The results suggest that for PEDV and TGEV Miller, APN may be the predominant receptor, while TGEV Purdue mainly uses an additional receptor for virus infection.

### 3.5. Effect of NA Treatment of Enterocytes and TGEV on Virus Replication

To assess the role of SAs as receptor determinants, enterocytes were pretreated with 50 mU/mL NA prior to inoculation with TGEV Miller and Purdue. Miller infected 1.0 ± 0.5% of the NA-treated enterocytes and 1.4 ± 0.4% of mock-treated cells. For Purdue, the percentage of infection was 2.3 ± 1.1% and 2.3 ± 0.9% for NA-treated and mock-treated cells, respectively. This implies that TGEV does not depend on terminal SA residues on the enterocytes surface for infection. Since it has been reported that removal of SAs on the surface of coronaviruses improves binding and infection [21], the replication of mock-treated, and NA-treated viruses was compared in untreated epithelial cells. NA pretreatment of virus significantly enhanced infection from 1.4 ± 0.4% to 2.9 ± 0.5% for Miller. For Purdue, NA pretreatment of virus significantly increased infection from 2.3 ± 1.1% to 7.9 ± 2.1% (Figure 5). These data show that removal of SA from TGEV promotes binding and replication of TGEV in enterocytes.

### 3.6. TGEV Infects Both SA Positive and Negative Enterocytes

Double immunofluorescence staining was further performed to assess the role of SA on TGEV replication. TGEV Miller could infect both SA positive cells (1.7 ± 0.3%) and SA negative cells (1.7 ± 0.2%). The TGEV Purdue replicated slightly better in SAs negative cells (4.5 ± 0.3%) compared to SAs positive cells (3.2 ± 0.8%; Figure 6).

### 3.7. TGEV binds to APN Positive/Negative and SA Positive/Negative Enterocytes

Primary enterocytes were inoculated with TGEV particles (m.o.i. = 10) at 4 °C. The binding of TGEV to APN positive/negative and SAs positive/negative cells was examined by double immunofluorescence staining. No significant differences were observed between the percentage of APN positive cells with bound TGEV Miller (2.3 ± 0.7%) and the percentage of APN negative cells with bound TGEV Miller (1.9 ± 0.2%; *p* = 0.49). The percentage of APN positive cells with bound TGEV Purdue (1.5 ± 0.7%) was lower than the percentage of APN negative cells with bound TGEV Purdue (3.9 ± 0.7%), but was not significantly different (*p* = 0.07). The percentage of Miller particles that colocalized with APN (65 ± 11%) was significantly higher than non-colocalized particles (35 ± 11%). The percentage of Purdue particles that colocalized with APN (33 ± 2%) was significantly lower than non-colocalized particles (67 ± 2%; Figure 7).

The percentage of SA negative cells with bound TGEV Miller (4.3 ± 0.3%) and with bound TGEV Purdue (5.0 ± 0.9%) was significantly higher than the percentage of SA positive cells with bound TGEV Miller (0.9 ± 0.4%) and with bound TGEV Purdue (2.1 ± 0.3%). The percentage of TGEV Miller particles (29 ± 13%) and TGEV Purdue (32 ± 12%) that colocalized with SA was significantly lower than the particles that did not colocalize, with Miller at 71 ± 13% and Purdue at 68 ± 12% (Figure 8).

## 4. Discussion

Porcine epithelial cells of the small intestines are the target cells for PEDV and TGEV in vivo. These cells show a high surface expression of APN, and APN has been reported to be the cellular receptor for PEDV and TGEV [7,9]. However, Vero cells with undetectable APN expression were historically used for PEDV propagation questioning the role of APN as a cellular receptor for PEDV [22]. In addition, overexpression of porcine APN in non-susceptible cells did not robustly support PEDV infection, and knock-out of APN in susceptible cells did not abrogate PEDV infection [13]. The present study was performed to determine the role of APN in PEDV and TGEV infection in their target primary porcine enterocytes. We found that a higher infection of PEDV and TGEV was correlated with a higher APN expression. However, both PEDV and TGEV did not only infect APN positive enterocytes, but also APN negative cells. Our results demonstrated that PEDV and TGEV may use another additional unknown receptor for entry in primary enterocytes.

The epithelium of the small intestines is continuously and rapidly renewed in a process involving cell generation and migration from the multi-potent stem cells in the crypts to the differentiated cells at the tips of the villi within 2–3 days. In our study, after three days cultivation, the primary enterocytes were positive for sucrase and iso-maltase, which are considered as differentiation markers for epithelial cells [19]. However, the expression of aminopeptidase N was only around 11%, indicating that the primary enterocytes are not fully mature at three days cultivation. Therefore, we analyzed APN expression in enterocytes at a later time point (seven days cultivation). A significantly higher expression of APN was detected at seven days cultivation compared to three days cultivation. Interestingly, the seven-day-cultured enterocytes with a higher APN expression showed significantly higher infection to TGEV than that of enterocytes cultured for three days. This agrees with the fact that the virus mainly infects and destroys mature enterocytes lining the villi of small intestines [23]. However, the infection efficiency of PEDV in intestinal epithelial cells was still low, indicating that other factors than APN need to be considered for PEDV infection. Therefore, several positive enterocyte differentiation factors were tested. Hydrocortisone plays an important role in the metabolism of proteins, lipids, and carbohydrates, and is a known promoter for differentiation of cultured cells. Hydrocortisone was found to be a critical factor for the differentiation of skeletal muscle, osteoblasts, and endothelial cells [24]. Sorrell and colleagues demonstrated that hydrocortisone significantly upregulated the expression of APN in human dermal fibroblasts [25]. Besides, Wnt signaling is required for the formation of normal crypt-villus units of intestines, and stimulates the differentiation of intestinal secretory epithelial cells [26]. Activated Wnt signaling has also been shown to promote mesenchymal differentiation [27]. The original rationale for including intestinal contents in our primary cell cultures was trying to mimic the in vivo situation in the intestinal tract. Intestinal contents contain a large number of enzymes, such as: Amylase, which digests carbohydrates to monosaccharides; pancreatic enzymes, which digest proteins into amino acids; and lipase which digests fats. It has been demonstrated that intestinal contents play an important role in virus infection. The proteases (trypsin) in the intestinal contents activate rotavirus infection by cleaving the outer capsid protein VP4 [28]. The propagation of porcine enteric calicivirus (PEC) on a cell line critically relies on the presence of intestinal contents in the culture medium [29]. Chang et al. demonstrated that the bile salts in intestinal contents are essential for growth of PEC by inducing the protein kinase A (PKA) signaling pathway [30]. In our study, the intestinal contents collected from the upper duodenum promoted the expression of APN and enhanced the infection of both TGEV and PEDV, especially the CV777 Vero adapted strain. Further investigation is needed to determine which growth-promoting factor in the intestinal contents is responsible for the increase of coronavirus infection.

To date, coronaviruses use four different proteins as cellular receptors. Mouse hepatitis virus (MHV) initiates the infection by binding to the carcinoembryonic cell adhesion molecule 1 on hepatocyte membranes and intestinal brush border membranes [31]. Next, APN was found to act as a receptor for porcine, feline, canine, and human coronaviruses [6]. The receptors for the highly pathogenic human respiratory viruses SARS-CoV type 1 and type 2 as well as Middle East respiratory syndrome coronavirus are angiotensin-converting enzyme 2 (ACE2) and dipeptidyl peptidase 4 (DPP 4) [32,33]. Cong and colleagues proved that the porcine small intestine epithelial cell line was more susceptible to PEDV when a high expression level of APN was present, and that interference with APN expression in epithelial cells inhibited PEDV infection, demonstrating that APN serves as an essential receptor for PEDV [12]. In addition, a transgenic mouse model expressing porcine APN was proven to be susceptible to PEDV, which confirmed that APN plays a role as the cellular receptor for PEDV [34]. In our study, we found that the higher infection of PEDV in enterocytes is correlated with higher APN expression. Both aged enterocytes and enterocytes treated with differentiation factors expressed more APN and were more susceptible to PEDV, which indicates that APN may play role in PEDV infection in enterocytes. Moreover, we found that the APN was expressed at the apical surface of enterocytes and PEDV infected 40 times more enterocytes at the apical surface than at the basolateral surface (Appendix A). These results were in agreement with the previous finding that PEDV enters polarized cells via the apical membrane [12]. Our results indicate that APN facilitates the entry of PEDV into primary enterocytes. Shirato and colleagues indicated that APN may promote PEDV replication in porcine kidney cell line, CPK cells via its protease activity [14]. How APN facilitates PEDV infection in enterocytes should be further investigated. However, due to the fact that other molecules besides APN will also be expressed at the apical plasma membrane during differentiation, they may also contribute to the higher susceptibility of differentiated enterocytes. Recently, increasingly more data has been published that show that APN is not a cellular receptor for PEDV. Overexpression of APN in non-susceptible cells did not confer susceptibility of the cells to PEDV and knocking out APN in susceptible cells did not abrogate PEDV infection, which indicates that APN is not required as a cellular receptor for PEDV in vitro [13,14,22]. Furthermore, APN knockout pigs retained their susceptibility to PEDV confirming that PEDV may use another additional receptor in pigs [35]. In agreement with these findings, we found that PEDV can infect APN negative primary enterocytes, which further confirmed that a cellular receptor different from APN exists in enterocytes for PEDV replication. The primary enterocytes isolated and cultured in our study are not 100% positive for APN, as we not only get the villi epithelial cells, but also the crypt epithelial cells during our isolation procedure. By immunofluorescence staining of ileum tissue of a three-day-old piglet, we observed that the villi epithelial cells are APN positive, while the crypt epithelial cells are APN negative (data not shown). As PEDV has been shown to infect goblet cells and crypt stem cells in addition to villous mature enterocytes [36], we believe that PEDV may also use another cellular receptor besides APN to infect intestinal cells. Taken together, our results indicate that although APN could significantly promote PEDV infection in enterocytes, an additional cellular receptor exists in enterocytes for PEDV replication. Since ACE2 is expressed in the gut epithelial cells, it will be tested in the near future if it can function as a PEDV receptor.

In addition to PEDV, APN has been identified as a major cellular receptor for TGEV. In the present study, it was shown that a higher expression of APN significantly increased the replication of TGEV in enterocytes, indicating that APN plays an important role as a cellular receptor for TGEV. Whitworth and colleagues demonstrated that APN knockout pigs were resistant to TGEV infection, indicating that APN is the sole functional receptor for TGEV [35]. However, we found that except APN positive enterocytes, TGEV could also infect APN negative enterocytes, suggesting that an additional receptor also exists in enterocytes for TGEV. The result obtained in our in vitro experiment may not fully reflect the in vivo experiments, as the in vivo situation is composed of a complex set of cells and tissues. The Purdue strain used in our study has been passaged 114 times in primary kidney cells and a nucleotide mutation (T to G at nucleotide position 1753), which causes a serine (S) to alanine (A) at aa 585 [37,38]. This mutation may be correlated with the cell adaptation and also it may result in a broader cell tropism of the adapted strain, which may explain that the virus is able to grow more efficiently in APN negative cells. Purdue has been proven to infect primary colon epithelial cells and porcine myofibroblasts, which are both negative for APN [19]. In vivo, TGEV shows a higher cell tropism to villous enterocytes of newborn piglets compared to older pigs and causes only high mortality in the early life of piglets. Since APN is present on enterocytes from both newborn and older pigs, the age-dependent susceptibility to TGEV infection may be caused by an additional receptor that is specifically present in newborn piglets. Taken together, we hypothesize that APN is the determinant cellular receptor for TGEV, but an additional receptor exists in young piglets. Apart from APN, TGEV also uses SA as an attachment mediator on the cells. Shahwan and colleagues have shown that NA treatment of jejunum epithelial cryosections did not reduce the TGEV Spike protein binding in vitro and were doubting on the role of SA during infection [39]. In our study, removal of SA from intestinal cells had no effect on TGEV infection, showing that terminal SA residues are not receptor determinants for TGEV. Removal of SA from TGEV virions significantly enhanced the viral infectivity in vitro. This indicates that SA on the virion surface masks the binding site of the viral protein to cellular receptors. Removal of SA from virions facilitates TGEV to bind to its functional receptor on the enterocyte membrane. To confirm the role of APN and SA in TGEV infection in primary enterocytes, a binding assay was performed in this study. The results demonstrated that both TGEV Miller and Purdue could bind to APN positive and SA positive cells. Meanwhile, Miller and Purdue could also bind to APN negative and SA negative enterocytes. Furthermore, our study showed that APN and SA double-negative enterocytes could be infected by TGEV Miller and Purdue (Appendix A), which suggests that besides APN and SA, TGEV can use another cellular receptor for the replication in enterocytes. Further investigation will focus on identifying this unknow receptor. A binding assay for PEDV could not be conducted due to the low titer of the virus stock.

## 5. Conclusions

Based on our previously established primary enterocyte co-culture system, which is very relevant for the in vivo situation, it was shown that a higher expression of APN on the enterocytes resulted in a higher infectivity of TGEV and PEDV. However, TGEV and PEDV could also infect APN negative enterocytes, indicating that an additional receptor exists in enterocytes besides APN. TGEV did not show binding activity on SAs on the surface of enterocytes. These new insights stimulate the search for unknown receptors for PEDV and TGEV, which can assist further research on antiviral intervention.

## Figures and Tables

**Figure 1 viruses-12-00402-f001:**
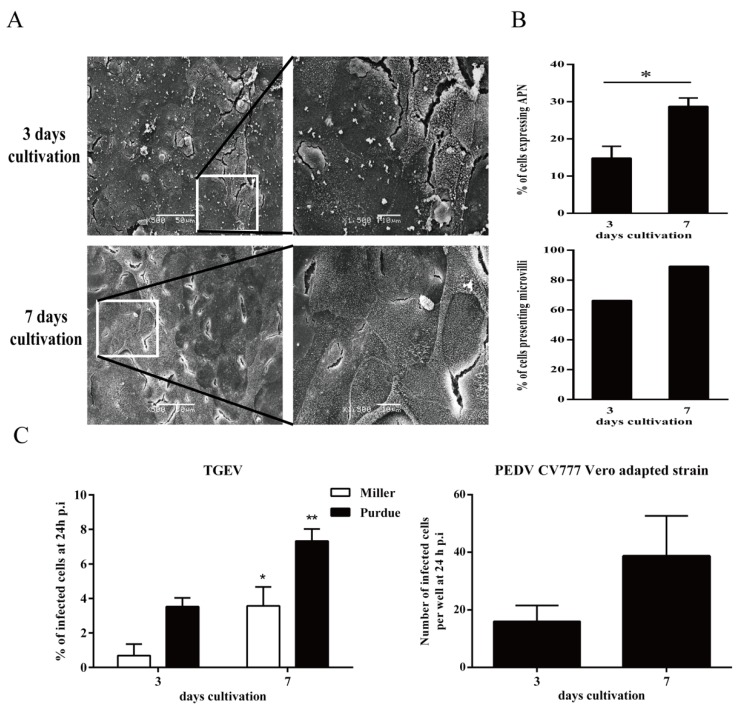
Effect of cultivation age on the percentage of microvilli positive cells and susceptibility to virus infection. (**A**) At three days and seven days cultivation, the microvilli on the surface of primary enterocytes were detected by scanning electron micrograph. (**B**) The percentages of microvilli positive cells and aminopeptidase N (APN) positive cells were counted. (**C**) At three days and seven days cultivation, cells were inoculated with Porcine epidemic diarrhea virus (PEDV; 10^5.6^ TCID_50_/mL) and transmissible gastroenteritis virus (TGEV; m.o.i. = 1). Percentage of infection was measured 24 h post inoculation. Data are expressed as mean ± standard deviation (SD) of the results of three separate experiments. Statistically significant differences in comparison with data from three days cultivation are presented as **p* < 0.05 or ***p* < 0.01.

**Figure 2 viruses-12-00402-f002:**
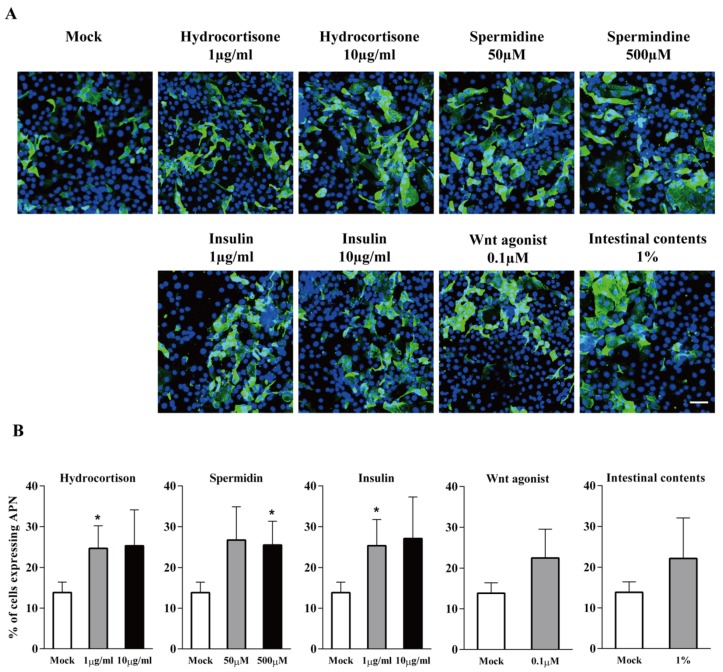
APN expression in primary ileum epithelial cells in the presence of enterocytes differentiation factors hydrocortisone, spermidine, porcine insulin, Wnt agonist, or small intestinal contents. (**A**) Immunofluorescence staining of APN expression (green) in enterocytes with different treatments. Scale bar: 50 µm. (**B**) The percentage of cells expressing APN at 24 h of treatment. Data are expressed as mean ± SD of the results of three separate experiments. Statistically significant differences in comparison with data from mock treatment are presented as **p* < 0.05.

**Figure 3 viruses-12-00402-f003:**
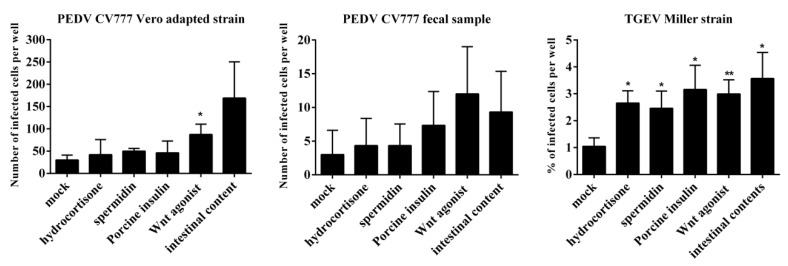
Infectivity of enterocytes to PEDV and TGEV after 24 h treatment with hydrocortisone, spermidine, porcine insulin, Wnt agonist, or small intestinal contents. Enterocytes were cultured with hydrocortisone, spermidine, porcine insulin, Wnt agonist or small intestinal contents. Twenty-four hours post cultivation, cells were inoculated with PEDV and TGEV. The level of infection was measured as number of infected cells per well (PEDV) or percentage of infected cells (TGEV) by immunofluorescence. Data are expressed as mean ± SD of the results of three separate experiments. Statistically significant differences in comparison with data from mock treatment are presented as **p* < 0.05 or ***p* < 0.01.

**Figure 4 viruses-12-00402-f004:**
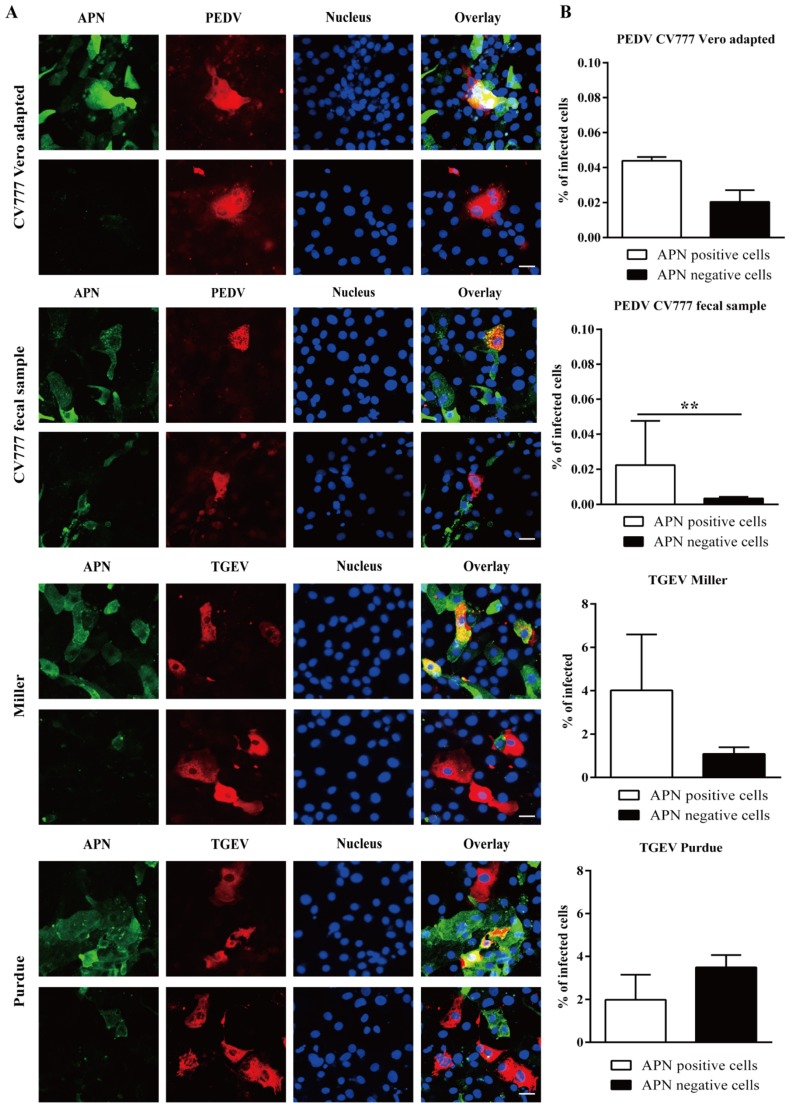
Infection of APN positive/negative enterocytes with PEDV and TGEV. (**A**) Double immunofluorescence staining of APN (green) and PEDV/TGEV (red) in primary enterocytes. Scale bar: 25 µm. (**B**) Percentage of infected cells within the population of APN positive and negative cells. Data are expressed as the mean ± SD of the results of three separate experiments. Statistically significant differences between APN positive and APN negative cells are presented as ***p* < 0.01.

**Figure 5 viruses-12-00402-f005:**
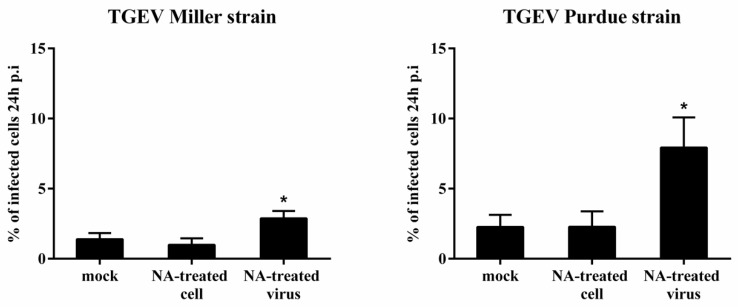
Effect of neuraminidase N (NA) treatment of enterocytes or virus on the infection of TGEV Miller and Purdue in enterocytes. The percentage of infection was evaluated 24 h post inoculation. Data are expressed as mean ± SD of the results of three separate experiments. Statistically significant differences in comparison with data from mock treatment are presented as **p* < 0.05.

**Figure 6 viruses-12-00402-f006:**
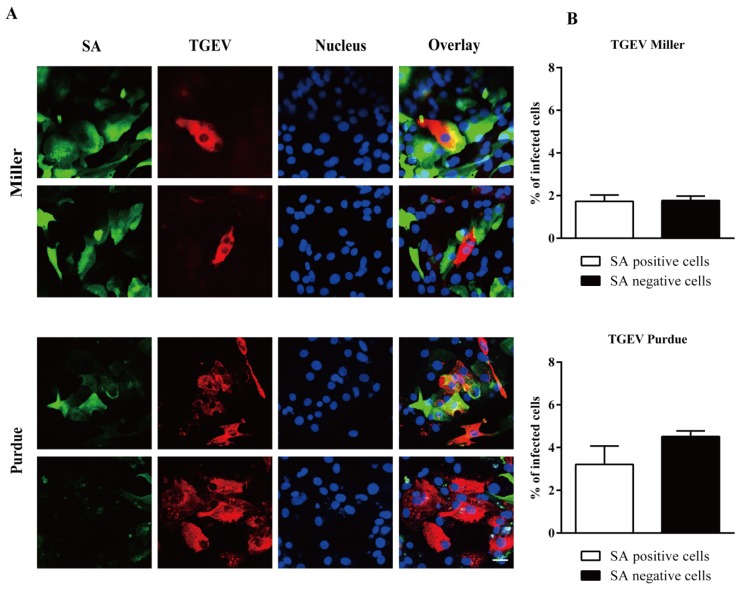
Infection of TGEV in sialic acid positive/negative enterocytes. (**A**) Double immunofluorescence staining of TGEV in primary enterocytes. Scale bar: 25 µm. (**B**) The percentage of infection among sialic acid positive and negative cells. Data are expressed as the mean ± SD of the results of three separate experiments.

**Figure 7 viruses-12-00402-f007:**
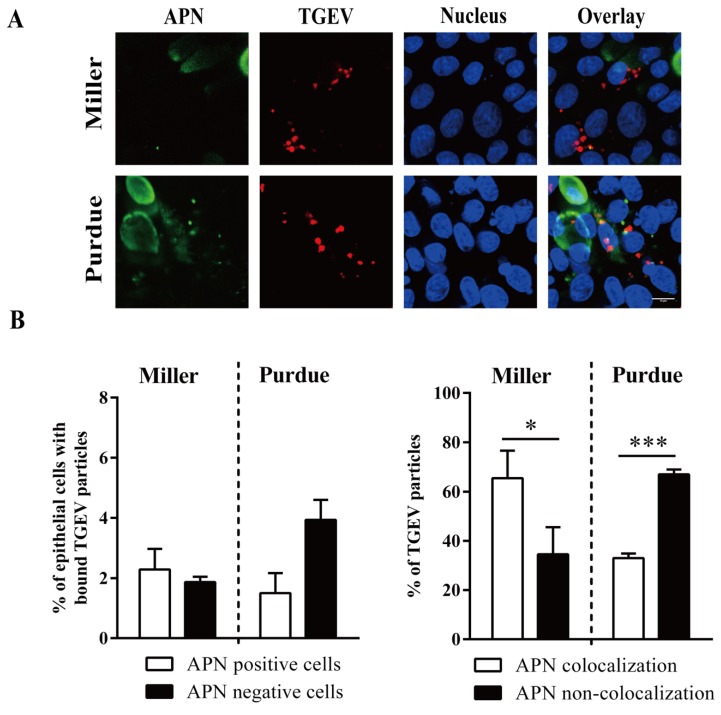
TGEV binds to APN positive/negative enterocytes. Primary enterocytes were inoculated at 4 °C with TGEV particles (m.o.i. = 10). (**A**) Double immunofluorescence staining of TGEV particles bound to APN positive/negative cells. Scale bar: 10 µm. (**B**) The percentage of cells with bound virus particles (left panel). The percentage of APN colocalized TGEV particles was counted based on five random fields (right panel). Data are expressed as the mean ± SD of the results of three separate experiments. Statistically significant differences are indicated as **p* < 0.05 and ****p* < 0.001.

**Figure 8 viruses-12-00402-f008:**
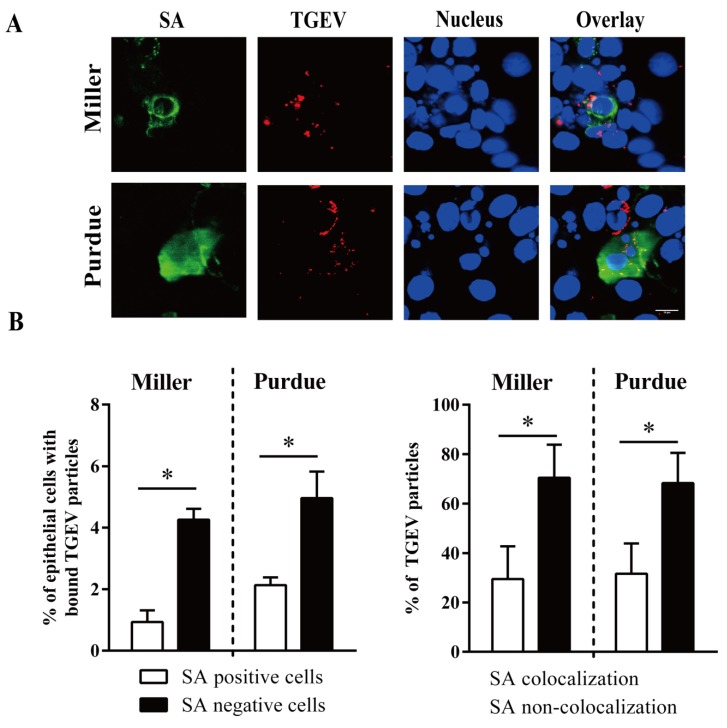
TGEV binds to sialic acid (SA) positive/negative enterocytes. Primary enterocytes were inoculated at 4 °C with TGEV particles (m.o.i. = 10). (**A**) Double immunofluorescence staining of TGEV particles bound to SA positive/negative cells. Scale bar: 10 µm. (**B**) The percentage of cells with bound virus particles (left panel). The percentage of SA colocalized TGEV particles was counted based on five random fields (right panel). Data are expressed as the mean ± SD of the results of three separate experiments. Statistically significant differences are indicated as **p* < 0.05.

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
