# Peer review of "Role of Porcine Aminopeptidase N and Sialic Acids in Porcine Coronavirus Infections in Primary Porcine Enterocytes"

_viruses, 2020, doi:10.3390/v12040402_

Round 1

Reviewer 1 Report

The authors investigated role of APN and SA as cellular receptors for PEDV and TGEV using primary procine enterocytes. Even though the results provided some insight on the role of APN and SA for the viral infection, I have several concerns on the methods and the results as follows;

Major)

  1. It seems that viral infectivity to the primary enterocytes was very low despite their expression of APN. The authors should characterize the primary cells they used before infection experiments. The relative percentile of enterocytes, microvilli-positive cells, and AP-positive cells should be carefully assessed by using specific markers and flow cytometry. In line 100-101, the commented co-culture of enterocytes with myofiboroblasts. Why did they co-culture them? Whole procedure for cellular preparation also needs to be described. I’d like to recommend to use pure culture of enterocytes and use APN-positive/negative cells after sorting them. Furthermore, they used multiple agents inducing cellular differentiation. They have to confirm cellular differentiation using specific markers before infection assay.
  2. In method section 2.5, two different methods for viral titration of PEDV and TGEV were used. They have to use the same method for relative comparison of viral infectivity. M.O.I. would be better for comparative analysis. In addition, Fig. 1C presented % of infected cells for TGEV and number of infected cells/well for PEDV. They have to use the same method for viral infectivity.
  3. Description on experimental methods was not clear for the preparation of several materials. For example, the preparation methods for small intestinal contents (line 106) and their composition should be presented. The methods for fecal suspension (line 103) and its viral titer needs to be presented. The sources of anti-viral antibodies (line 147-148) needs to be described.

Minor)

  1. The author suggested removal of SA from virion surface enhance the infection of TEGV. They have to confirm the level of SA in the virions by quantitative analysis (i.e. WB) and present the data to ensure its effect on viral infectivity.
  2. Section 3.1. Aged enterocytes…enterocytes were from 3 days old piglets. It’s not aged cells.
  3. The viral infectivity is generally very low. It’s better to use higher viral moi for infection.
  4. line 357, primary enterocytes were positive for sucrose and iso-maltase… The authors should present the data
  5. line 412 – 414. Data on APN expression at the apical surface needs to be presented.
  6. IACUC permission is required for the use of piglets.

Reviewer 2 Report

This study is very interesting, to reveal the infection of PED and TGE. And the quality of this study is so high and the methods and the results are so clear. I hope that the mechanisms of APN related infection in coronavirus, would reveal the infection mechanisms of Covid19 and produce new drugs against coronavirus infection. The manuscript is written by clear English, but, I hope that abbreviations as bellows show on a list for clear understanding of this study.

APN, Wnt agonist, NA, SA

Reviewer 3 Report

The manuscript by Cui et al describes the study on receptors used by PEDV and TGEV, two coronaviruses infecting pigs. The study is done in primary enterocytes and that is the attractive part of the study. Staining for aminopeptidase N (APN) is done to determine whether the viruses can also enter the cells when the receptor is not expressed. Furthermore, treatment with neuraminidase is used to investigate the influence of sialic acids on the cell surfaces or on the virus. An attractive part of the manuscript is the treating of the cells with intestinal content, to let the cells differentiate by the most natural stimulants. The study is of high relevance, as these porcine viruses are still circulating with sometimes devastating disease in farm outbreaks. Although experiments are sound, there is a problem with drawing conclusions. Probably due to high variation among duplicates some comparisons between conditions in an experiment do not reach statistical significance, yet they are referred to as “higher” “lower” “increase” or “decrease”.

Major comments:

  • Results: The authors should make a choice: either only show and discuss the data for which significant difference (P< 0.05) is measured, or describe all data, as it is done now, yet indicate for all not statistically significant measures that there is a trend, but not reaching statistical significance (with all non-significant p-values added in the text) and mention why it is not significant where relevant (so mention for instance if this was because of large variation among tests, or because the trend is there but differences are small and number of replicates also only a few, and for that reason not significant).
  • The conclusion in line 472-473, where it is mentioned that higher expression of APN correlated with higher infectivity of PEDV can not be concluded. It is only shown that a higher degree of differentiation correlates with more cells being infected if I read the paper right. APN expression increases with differentiation, but there are many surface molecules increased when cells are more differentiated.
  • It may very well be that APN is a receptor for TGEV and PEDV, but I think that the manuscript is most interesting in showing that this molecule is NOT essential, and that there is very likely another cellular surface molecule that can be used for entry. That this is the most intriguing finding is not so clear from the discussion.
  • I miss in the discussion the option that the unknown receptor that facilitates entry in APN negative cells might be ACE2. It is known that HCoV-NL63 uses ACE2 to enter the target cells, and HCoV-NL63 is like TGEV and PEDV an alpha coronavirus. How likely is it that this is the mystery extra receptor on primary cells? Or has this been investigated already? If so, please add that in the discussion.
  • Discussion is a bit lengthy and in some parts a repetition of the results. Please focus only on: APN is not always the receptor for these viruses. What could be the effect of lab-adaptation on APN usage. What is known from literature on APN being essential, was this all with lab adapted strains? What could be the most likely candidate for an alternative receptor.
  • Secondly, it would be nice to add in the discussion what the authors advice for culture. What is the best stimulant that enhances primary enterocyte differentiation?
  • Line 400 and 401-403. Add in sentence 400 “type 1 and type 2” after the SARS-CoV, and delete SARS-CoV-2 info in line 401-403, it is an unnecessary addition.

Minor comments:

  • Abstract line 14 and 15. Mention the percentages of cells with microvilli, as “more” is not very scientific. Also mention the fold rise in infectivity instead of “more”
  • Introduction line 45. What is the relevance of mentioning the human mAb directed towards S1 of SARS-CoV?
  • Please describe in the M&M how the 1% solution of small intestine content was made. I assume a filtration was done to remove bacteria?

Round 2

Reviewer 1 Report

The revised MS is improved well enough to be published in Viruses.